# WS_2_ Nanosheet Loaded Silicon-Oxycarbide Electrode for Sodium and Potassium Batteries

**DOI:** 10.3390/nano12234185

**Published:** 2022-11-25

**Authors:** Sonjoy Dey, Gurpreet Singh

**Affiliations:** Department of Mechanical and Nuclear Engineering, Kansas State University, Manhattan, KS 66506, USA

**Keywords:** sodium-ion battery, potassium-ion battery, TMD, WS_2_, tungsten disulfide, PDC, SiOC, ceramics

## Abstract

Transition metal dichalcogenides (TMDs) such as the WS_2_ have been widely studied as potential electrode materials for lithium-ion batteries (LIB) owing to TMDs’ layered morphology and reversible conversion reaction with the alkali metals between 0 to 2 V (v/s Li/Li^+^) potentials. However, works involving TMD materials as electrodes for sodium- (NIBs) and potassium-ion batteries (KIBs) are relatively few, mainly due to poor electrode performance arising from significant volume changes and pulverization by the larger size alkali-metal ions. Here, we show that Na^+^ and K^+^ cyclability in WS_2_ TMD is improved by introducing WS_2_ nanosheets in a chemically and mechanically robust matrix comprising precursor-derived ceramic (PDC) silicon oxycarbide (SiOC) material. The WS_2_/SiOC composite in fibermat morphology was achieved via electrospinning followed by thermolysis of a polymer solution consisting of a polysiloxane (precursor to SiOC) dispersed with exfoliated WS_2_ nanosheets. The composite electrode was successfully tested in Na-ion and K-ion half-cells as a working electrode, which rendered the first cycle charge capacity of 474.88 mAh g^−1^ and 218.91 mAh g^−1^, respectively. The synergistic effect of the composite electrode leads to higher capacity and improved coulombic efficiency compared to the neat WS_2_ and neat SiOC materials in these cells.

## 1. Introduction

With innumerable advantages such as high round trip efficiency, long cycle life, and minimal maintenance of electrochemical energy storage, rechargeable batteries have opened the door to innovative technologies such as mobile electronics and electrified transportation [1]. Furthermore, the widespread application of electronic devices in the healthcare, fitness, sensing, sports, and textile industries has shifted consumer focus to flexible and stretchable electronics [2]. Consequently, rechargeable batteries, specifically LIB technology, have drastically developed in the last two decades. Although LIBs are superior in the electrochemical energy storage systems (EESS) due to their high energy density and lightweight characteristics, their questionable sustainability, low theoretical capacity (theoretical specific capacity of ~372 mAh g^−1^) of anode material in advanced LIBs (does not match the energy density requirement), and high cost of lithium metal and other battery components have shifted the research attention to other options. Thus, high-capacity batteries based on lighter alkali metals, such as sodium (Na) and potassium (K), are becoming valid contenders due to their abundance and minimal expense [3]. However, Na and K ions have larger ionic radiuses (1.02 Å for Na^+^; 1.38 Å for K^+^), rendering graphite-based anodes with an interlayer spacing of 3.35 Å useless. Also, high volume expansion (61% for K^+^ ions) causes structural disintegration and subsequent capacity fading in graphite during charge and discharge cycles [4,5].

When graphene was first exfoliated from graphite using the mechanical cleavage technique, the scientific community’s elevated interest was focused on developing similar two-dimensional (2D) materials. 2D TMDs contain a sheet-like graphene morphology with hexagonally packed layers of transition metal atoms sandwiched between two chalcogen atoms. Although the intralayer metal chalcogen bonds in TMDs are covalent, the sandwich layers are coupled by van der Waals bonds which are typically weak, rendering them useful for application in vast fundamental and technological disciplines such as catalysis, energy storage, sensing, and electronic devices [6]. Among the various 2D TMDs, WS_2_ contains a lamellar structure with an interlayer spacing of 0.62 nm between (002) lattice planes, thus offering suitable space for the reversible insertion of large-sized Na and K ions [7]. Moreover, because WS_2_ has a superior volumetric energy density of 3248 mAh cm^−3^ as a LIB electrode due to the high density (>7 g cm^−3^), WS_2_ is an auspicious choice for anode material [8]. Although significant research has investigated WS_2_ as a potential lithium-ion storage application, the application of WS_2_ in NIBs and KIBs is somewhat limited. Unique, distinctively tailored nanostructures of WS_2_, such as nanoflakes, nanosheets, nanowires, and nanoflowers, have also been studied to determine if they enhance the performances of NIBs [9,10,11,12,13,14]. Still, these require extensive conditions to fabricate such structures. On the other hand, when commercial WS_2_ powder was investigated for K-ion storage behavior, the capacity was only 103 mAh g^−1^ in a potential window of 0.01–3.0 V at 100 mA g^−1^ [15]. Furthermore, few investigations have found that when alkali metal ions are inserted and extracted within the layers, the failure of the electrode structure occurs as it goes through significant volume change induced by mechanical stress. As a result, the delamination between the active material and the current collector results in poor stability over a more extended period [16,17,18]. Moreover, the widely studied TMD materials prefer to undergo irreversible conversion reactions rather than reversible intercalation reactions when interacting with alkali metal ions forming an inert layer and causing first-cycle losses [19]. The layer formation is associated with the dissolution of polysulfides in the electrolyte used in the device. With issues mentioned above, further investigation is needed to determine a simple and cost-effective fabrication process to enhance electrode capacity in NIBs and KIBs.

To overcome the difficulties discussed above, when the TMD materials as battery electrodes, are amalgamated with carbon materials, metal sulfides, oxides, and conducting polymers which impart electronic conductivity within the composite material and boosts the electrochemical properties [20]. With unique physical and chemical properties, PDC materials are also eying candidacy for their use in electrochemical energy storage devices. When the microstructure of PDC is revealed, a network of sp^2^-bonded carbon atoms can be located at the tetrahedral silica boundary. In addition, the Si-C bonds are formed in PDCs, and these bonds, along with the free carbon network resembling graphene, prevent the diffusion of chemicals and degradation. For example, Davi et al. have shown that this amorphous structure of PDCs helps protect MoS_2_ from degradation and prevents the loss of sulfur species in PDC/MoS_2_ composite, thereby improving capacity retention in NIB electrodes [18]. However, the use of PDCs as KIB electrodes are yet to be thoroughly studied, which prompted the current investigation.

The present study sought to chemically functionalize WS_2_ nanosheets with PDCs by fabricating a fiber structure via electrospinning, thus obtaining electrodes with lucrative electrochemical properties. The fibermat electrode was achieved by preparing a spinnable solution with liquid phase preceramic oligomer and exfoliated WS_2_ nanosheets, followed by pyrolysis. The obtained non-woven fibermat structure contained an arid amount of WS_2_ nanosheet embedded into the SiOC structure that helped buffer the volume change induced by the TMD material when used as Na^+^ ion and K^+^ ion half-cells. The microscopic and spectroscopic analysis confirmed the presence of WS_2_ nanosheets and the SiOC structure, exhibiting improved rate capability compared to the neat materials in NIBs and KIBs. This is potentially the first investigation to feature WS_2_ nanosheet and PDC in fibermat form for use beyond LIB chemistry.

## 2. Materials and Methods

### 2.1. Electrospinning Solution Preparation

The neat WS_2_ powder utilized in this study was purchased from Alfa Aesar (Haverhill, MA, USA). The preceramic silicon oligomer 1, 3, 5, 7-tetramethyl, 1, 3, 5, 7-tetravinyl cyclotetrasiloxane (4-TTCS) was purchased from Gelest (Morrisville, PA, USA) and used without further modification. The solution preparation process for the electrospinning process included sonicating the obtained WS_2_ powder with isopropyl alcohol purchased from Fisher Chemical (Lenexa, KS, USA) for 1 h in a probe sonicator. The result was a black dispersion. This 4-TTCS and Polyvinylpyrrolidone (PVP) with an average molecular mass of 1,300,000 g mol^−1^ purchased from Sigma Aldrich (Missouri, MA, USA) were mixed with the previous solution via stirring until a homogenous mixture was obtained. The weight ratio of the 4TTCS: IPA was kept at 1:9, and the weight ratio of the WS_2_ powder: 4TTCS was 1:5. PVP was mixed at a weight ratio of 1:3 with 4TTCS to induce viscosity to the overall solution as the preceramic oligomer possessed low molecular weight (<500 g mol^−1^). A small amount of Dicumyl peroxide (1 wt. % of 4TTCS) purchased from Sigma Aldrich was also mixed with this solution which worked as a cross-linking agent.

### 2.2. Electrospinning Conditions and Fibermat Fabrication

Preparation of fibermat containing WS_2_ nanosheets is a multistep process in which the first step is to produce a flowable and viscous solution (>10 poise) favorable for spinning conditions and as described in the previous section. The as-prepared solution was loaded into a syringe with a metallic needle connected with a high-power voltage source grounded with a roller collector. Aside from the prepared solution, the spinnability of fibers depends on various other parameters. Examples include the feed rate of the solution (kept at 5 mL h^−1^), optimization of the applied electric field between the needle and the roller collector (19 kV was utilized in this study), and the distance between the needle tip and the roller collector (constantly kept at 12 cm). Such conditions enabled us to fabricate the as-spun fibermat about 20 × 20 cm^2^ in area and dark greyish in appearance. The as-spun fibermat was kept in a low-temperature oven for 6 h at a temperature of 160 °C to cross-link under ambient conditions. A small portion of this fibermat (30 × 25 mm^2^) was taken into a ceramic boat and pyrolyzed at higher temperatures in a tube furnace. The heat treatment was carried out in a two-step process at 400 °C for 1 h and 800 °C for 30 min under an inert atmosphere (Argon gas). The heating rate was kept constant at 2 °C per minute. Thus, the final pyrolyzed WS_2_/SiOC fibermat was obtained.

### 2.3. Characterization Techniques

The morphology of the fibermat was investigated with two SEM microscopes, namely Carl Zeiss EVO MA10 and FEI Nova NanoSEM450, with a 5–30 kV impinging voltage. Transmission electron microscopic images were obtained using Phillips CM100 TEM (Nashville, TN, USA) under an accelerating voltage of 100 kV. The chemical analysis of the samples was done by X-ray photoelectron spectroscopy (XPS), obtained using PHI Quantera SXM using monochromatic Al-Kα with an energy of 1486.6 eV. The Raman imaging system of Horiba Jobin Yvon LabRam Aramis was utilized using a He-Ne laser (wavelength of 633 nm and power of 17 mW) to collect the Raman spectra of the pyrolyzed samples. The FTIR spectra of the fibermats were collected utilizing the Spectrum 400 FT-IR spectrometer from PerkinElmer (Waltham, MA, USA).

### 2.4. Electrochemical Analysis

The electrodes studied in this work were prepared in the following composition: 70 wt.% of active material (WS_2_/SiOC pyrolyzed fibermats), 15 wt. % of carbon black as the conducting agent (Alfa Aesar, Haverhill, MA, USA), and 15 wt. % of polyvinylidene fluoride as the binder (Alfa Aesar). A homogenous slurry was obtained upon mixing the materials mentioned above by adding 1-Methyl-2-pyrrolidinone (Sigma Aldrich, Missouri, MA, USA) in a dropwise manner. This slurry was then coated on a 9 µm thick copper substrate and then dried at 80 °C for 18 h to remove solvents. The thickness of the then-obtained film was found to be 125 µm. Circular sections were obtained with a 14.29 mm diameter circular punch to be used as a working electrode in the 2032 type coin cell. The active material (WS_2_/SiOC) mass loading for the NIB and KIB cells was 0.63 mg each. The negative electrode in the case of NIBs was sodium metal with 1.0 M NaClO_4_ (Alfa Aesar) in (1:1 *v*/*v*) EC: DMC (anhydrous, 99%, Sigma Aldrich) as electrolyte. On the other hand, the negative electrode in the case of KIBs was potassium metal with 1.0 M KPF_6_ (Alfa Aesar) in (1:1 *v*/*v*) EC: DMC (anhydrous, 99%, Sigma Aldrich) as electrolyte. In both cases, the two electrodes were separated by a monolayer membrane (Celgard), and the glass separator was soaked with electrolytes. All the cells were assembled in a high-precision argon atmosphere with O_2_ and H_2_O contents below 0.1 ppm.

## 3. Results and Discussion

### 3.1. Microscopic Analysis

As the fibermats were first electrospun and then cross-linked and pyrolyzed at elevated temperatures, it was essential to investigate the surface of the fibers as the effect of temperature could be diverse. Thus, SEM micrographs were obtained for the as-spun (Figure 1a), cross-linked (Figure 1b), and pyrolyzed fibermats (Figure 1c). The inset of each figure shows distribution plots obtained from ImageJ software to study fiber diameters. The average diameter for the as-spun fibers was 6–7 µm, while the average diameters for the cross-linked fibers were 3–4 µm, and the diameter range of the pyrolyzed fibers was 1.6–2.0 µm. Magnified SEM micrographs (Figure 1d,e) of the final pyrolyzed fibermat showed regular fibers with no anomalies such as beads or defects. The WS_2_ nanosheets were sporadically scattered as islands in the fiber geometry. The fiber diameter decreased as a function of the temperature and was further investigated via spectroscopic methods. The change in fiber diameter in different conditions was comparable to the weight-loss phenomenon recorded in various annealing stages, as shown in Appendix A–c. A digital camera image showed a gray WS_2_/SiOC as-spun fibermat, which correlated to the color of the near WS_2_ fibers but differed from the neat SiOC fibermat. The cross-linked fibermat, however, had a light gray appearance with a weight retention of 56.04% and a linear shrinkage of 4.9%. The final pyrolyzed fibermat was black, with a weight retention of 52.27% of the cross-linked fibermat. Linear shrinkage was observed at 51.43%. The weight loss and linear shrinkage records were then corroborated by the fiber diameter from the SEM images. The elemental distribution map obtained from the Energy Dispersive Spectroscopy (EDS) showed an even distribution of the materials in the fibermats. The W-L*α*-1 and S-K*α*-1 maps in Figure 1f,g show the nanosheet-like distribution for these elements in the fibermat, implying that the WS_2_ nanosheets are well bonded with the SiOC fiber matrix. Si-K*α*-1, C-K*α*-1,2, and O-K*α*-1 maps in Figure 1h–j show that these elements were distributed in a fiber-like fashion. A qualitative analysis of the elements in the fibermats was also carried out via the EDS, shown in Figure 1k. Characteristic W-L*α* at (8.6 keV) and S-K*α* (at 2.52 keV) spectral lines were observed in the fibermat, confirming the presence of WS_2_ nanosheet material. In addition to the peaks, spectral lines for Si-K*α* (at 1.97 keV), O-K*α* (at 0.73 keV), and C-K*α* (at 0.48 keV) were also observed, denoting the presence of Si-O-C matrix after pyrolysis. Notably, calibration caused the entire spectra to shift by ~0.2 keV. 

TEM was conducted to gain further insights into fiber morphologies. Bulk WS_2_ powders were first studied (Figure 1l) to confirm the layered morphology of the crystal. Figure 1l shows layered WS_2_ sheets with edge angles of 120°. The crystalline nature of this sheet was revealed by the selected area electron diffraction (SAED) pattern, denoting the 100 and 110 crystal planes [7,21] (shown in Figure 1m). The magnified TEM image of the WS_2_ nanosheet containing SiOC fibers is illustrated in Figure 1n, which shows the numerous pores through which the electron beam could pass. These pores are likely being generated due to gas evolution during polymer-to-ceramic transformation at elevated temperatures. Liwen et al. have demonstrated that these pores are beneficial for electrochemical storage applications as the porosity induces the charge transfer at the electrode-electrolyte interface and attenuates the volume change providing a feasible container space [22]. The SAED pattern taken from the WS_2_ nanosheet within the fiber shows the same hexagonal spotted pattern observed from Figure 1o with 100 and 110 crystal planes indicating that the WS_2_ nanosheet retains its crystalline nature even after pyrolysis. The micrographs show that the WS_2_ nanosheets maintain their crystalline structure even after functionalizing with SiOC material in fiber form. This formation might lessen the polysulfide dissolution when cycled in alkali metal-ion batteries. As the fibermats were first electrospun and then cross-linked and pyrolyzed at elevated temperatures, it was essential to investigate the surface of the fibers as the effect of temperature could be diverse.

### 3.2. Spectroscopic Analysis

Raman Spectroscopy, an all-embracing technique, was used to decipher the microstructure of the fibermat and designate the free carbon phase and the presence of the WS_2_ nanosheet materials. Figure 2a illustrates the Raman spectra of the bulk WS_2_ powder that can be compared with the WS_2_ nanosheet within the fibermat geometry. The figure also shows many major and minor peaks in the spectrum. Minor peaks with low intensity in the spectrum include A_1g_(M)-LA(M) at 230 cm^−1^, 2LA(M)-3E^2^_2g_(M) at 264 cm^−1^, 2LA(M)-2E^2^_2g_(M) at 297 cm^−1^, and 2LA(M) at 350 cm^−1^. In addition, the spectrum shows two major high-intensity peaks: first-order optical modes E^1^_2g_ at 354 cm^−1^ and A_1g_ at 418 cm^−1^_._ Notably, the 2LA(M) mode overlapped with the E^1^_2g_ mode. While in backscattering geometry, the first-order modes E^1^_2g_ and A_1g_ were prominent, with the Brillouin zone edge mode primarily being activated by disorder-induced within the sample, thereby identifying the longitudinal acoustic mode at the M point, namely LA(M). Specifically, the LA(M) mode results from the in-plane collective movement of atoms within the lattice, identical to sound waves. The atoms periodically compress and expand to create such waves along the propagation direction. Additional peaks in the spectra were the result of multi-phonon combinations of the above mode [23,24,25,26,27]. Although the LA(M) mode at 176 cm^−1^ was shallow in this investigation, the 2LA(M) mode combined with the E^1^_2g_ to render a high-intensity peak. All the first-order and second-order peaks occurred in the WS_2_ bulk material and WS_2_ nanosheets in the fibermat. For the WS_2_/SiOC fibermat sample (shown in Figure 2b), the high-intensity peaks were E^1^_2g_ and A_1g_ bands at 350 cm^−1^ and 417 cm^−1^, respectively. 

Another experimental feature from the WS_2_/SiOC sample’s Raman spectra was the filler material’s layer thickness. Because the exfoliated WS_2_ nanosheets were used to prepare the electrospinning solution, some features of the few layers of thick nanosheets were identified as expected. In summary, the A_1g_ peaks underwent a slight red shift. In contrast, the E^1^_2g_ peak experienced a subtle blue shift, as indicated by the few-layer nature of the nanosheets, potentially due to the sonication of the WS_2_ nanosheet material during solution preparation for the as-spun fibermat [28]. Moreover, the intensity ratio of the E^1^_2g_ to A_1g_ peak of the WS_2_ nanosheets in the fibermat was 0.83, indicating a few layered natures of the nanosheets within the fibermat. However, the intensity ratio of the two major peaks of the bulk material was in the range of 0.48, which accurately reflects previous studies by Zeng et Al. and Berkdemir et Al. [23,29].

Peak-fitted Raman spectra from Figure 2c showed the free-carbon phase of the WS_2_/SiOC fibermat. The prominent D and G peaks occurred at 1351.65 cm^−1^ and 1590.17 cm^−1^, respectively. While the D peak was due to disordered carbon in the sample, the G band (E_2g_ symmetry) caused in-plane stretching of the sp^2^ carbon bonds [30,31]. The deconvoluted Raman spectra revealed T and D″ peaks in the shoulder of D and G bands at 1246.22 cm^−1^ and 1481.68 cm^−1^, respectively. The T band signified the sp^2^–sp^3^ carbon-carbon double and triple bonds. The D″ band signified the amorphous nature of carbon in the sample, which was verified in TEM diffraction pattern studies [32]. The high degree of disorder present in the fibermats could also be verified from the intensity ratio of the D and G bands. For instance, I_D_/I_G_ ratio for the WS_2_/SiOC sample was 0.71.

Fourier transform infrared spectroscopy (FTIR) was used to identify the polymerization within the fiber geometry during the decomposition stages. Figure 2d shows FTIR spectra of the as-spun WS_2_/SiOC fibermat. In general, peaks observed in the region of 1265–1285 cm^−1^ were responsible for the -C-N bond originating from the PVP molecule. In addition, peaks in the 1650–1660 cm^−1^ region caused the carbonyl group in the molecule. The two peaks mentioned above in the functional group and fingerprint regions are primary indicators of the presence of the PVP molecule in the as-spun fibermat [33]. FTIR spectra also verified the facile integration of WS_2_ nanosheets with carbon-based groups. In addition, the peaks at ~650 cm^−1^ and ~1080 cm^−1^ were responsible for the W-S and S-S bonds, respectively [34,35]. Multiple peaks at 650 cm^−1^ to 800 cm^−1^ refer to the samples’ C-S bonds. For example, the peak at 790 cm^−1^ is characteristic of C-S-S-C, H-C-S, and C-S bonds [36,37]. Furthermore, the strong absorption from 1000 cm^−1^ to 1070 cm^−1^ indicates the Si-O bonds. The presence of the -CH_3_ deformation and asymmetric stretching in Si-CH_3_ bonds is visible in the range ~1261 cm^−1^ and 2961 cm^−1^ region, respectively. The absorption peak at 800 cm^−1^ indicates the Si-C stretching vibration [38]. These peaks confirm the presence of preceramic polymer in the fiber morphology by previous investigations [39,40]. For the cross-linked fibermats (Figure 2d; spectra in the middle), peak intensity decreased for both fibermat samples, indicating the occurrence of vinyl polymerization. Finally, the pyrolyzed fibermats (Figure 2d; spectra at the top) show Si-O and Si-C bonds.

In this study, XPS, an important technique used in the chemical analysis field for analyzing surface characteristics, was utilized to provide details regarding elements that compose the material and the chemical and electronic state. Figure 3a represents the XPS survey scan of the WS_2_/SiOC fibermats. The presence of W, S, Si, O, and C can be verified from the survey scan of the fibermats with no contamination with other materials within the instrument sensitivity. Appendix A presents the atomic percentage of the elements in the fibermat. WS_2_ in the form of nanosheet is found at 1.5% for W and 1.19% for S in the fibermat. Moreover, the high amount of carbon in the fibermat (73.42% found from the survey scan) also confirms the free carbon phase after the pyrolysis cycle, which was previously proven by the Raman spectroscopic analysis. Figure 3b. illustrates the high-resolution W4f and S2p spectra of the fibermats, confirming the bonding states of W present in the fibermat. Tougard function was used to subtract the background, and the spectra were fitted using the Voigt function. Spin-orbit splitting (s.o.s.) produced two core level peaks for each W4f and S2p, namely 4f7/2, 4f5/2, 2p3/2, and 2p1/2, respectively. For example, the binding energy of W4f showed that the W^4+^ state presented 4f7/2 and 4f5/2 bands in 33.6 eV and 31.6 eV, respectively. The XPS spectra efficiently captured the effect of sonication of the bulk WS_2_ powder when preparing the electrospinning solution. The sonication caused the formation of 2H and 1T/1T’form, as shown by the abundant W4f7/2 component in the sample due to high peak intensity. Slight oxidation of the layered WS_2_ nanosheets was also observed with characteristic peaks of WO_x_ at 35.4 eV and 37.5 eV. The presence of the WO_x_ component may result from oxygen functionalities originating from the preceramic polymer. 

The high-resolution S2p spectra can also be deconvoluted into two components: 2p1/2 and 2p3/2 at 163.1 and 161.8 eV, respectively. These intermediate doublets were due to sulfur S^2−^ ions in WS_2,_ as noted in previous investigations of WS_2_ nanosheets and nanoflakes [41,42,43]. The high-resolution spectra of the Si2p, O1s, and C1s were also obtained for the fibermat and shown in Figure 3c–e, respectively. The deconvoluted Si2p spectra represented the Si-O and Si-C bonds present in the fibermat. From the C1s spectra, the C-Si, C-C, and C=O bonds were identified. From the O1s spectra of the fibermat, the Si-O and, SiO_2_ bond states were observed with a dominance of the SiO_2_ over the other. From all this information, it is apparent that a good amount of carbon in the sample is also bonded with Si, and individual SiO_2_ regions were observable, which validates the SiOC microstructure of the pyrolyzed fibermat even after pyrolysis while including WS_2_ in the nanosheet form [44,45].

### 3.3. Electrochemical Analysis

An investigation of Na^+^ ion and K^+^ ion electrochemical reactions at various potentials was carried out using a galvanostatic charge–discharge (GCD) profile. The differential capacity curves obtained from the GCD profile provide a clearer view of the plateaus available in the GCD profile, converting them into individual peaks. Appendix A depicts the GCD profile and corresponding differential capacity curve of the bulk WS_2_ nanosheet material when used as an anode in the Na-ion half-cell. From the differential capacity curve of the Na-ion half-cell (shown in Appendix A), it is apparent that the Na-ion insertion process in the bulk material is multistep, with several peaks appearing in the charge and discharge cycles. In the first sodiation process, two major peaks at 0.29 and 0.6 V (vs. Na/Na^+^) can be attributed to the irreversible conversion reaction of WS_2_ with sodium ions into metallic nanoparticles embedded into the Na_2_S matrix and the formation of solid electrolyte interphase (SEI) layer on the surface of the electrode [46]. Again, during the 1st desodiation cycle, oxidation of W to WS_2_ is located at the curve in regions 1.89 and 2.3 V (vs. Na/Na^+^) [26,47]. In the second and third cycles, the peaks at 0.29 and 0.60 V diminish, indicating that the reaction is irreversible. The reversible reaction between Na^+^ ions and WS_2_ can be traced to the peak at 1.36 V (vs. Na/Na^+^), which is apparent only from the second cycle onwards [26]. For the desodiation process in the 2nd and 3rd cycles, the before-mentioned peaks seem to have weakened and broadened. From the GCD curve denoted by Appendix A, the first cycle charge capacity of the WS_2_ neat electrode is found to be 380.42 mAh g^−1^ with a coulombic efficiency of 67.2% at a current density value of 100 mA g^−1^. This initial irreversible capacity loss is thought to be originating from the conversion reaction and SEI layer formation during the first stages of cycling, which is a typical case for layered compounds. Achieving stability in this layer formation takes some time for a neat WS_2_ electrode, so the high capacity drop in the first five cycles is observed. The charge capacity achieved stability after the first five cycles. In a rate capability test (where current density was increased orderly), the material retained its stable coulombic efficiency only after the 10th cycle, reaching nearly 100% and maintaining a similar value till the 45th cycle. In the rate capability test (shown in Figure 4c), the capacity retention during the 26th, 31st, 36th, and 41st when the current density was brought back to the initial rate (600, 400, 200, 100 mA g^−1^, respectively) was found out to be 98.7%, 92.4%, 81.78%, 59.54%, respectively. 

Although silicon-based materials are thought to be almost inactive in NIBs, recent investigations report that free-carbon domains existing in such materials are well suited to store Na^+^ ions. Therefore, it is logical to expect Na^+^ ion storage property originating from the existence of free carbon domains, which is evident in the spectroscopic analysis of the SiOC material used in this study. Neat SiOC fibermats were studied in GCD experiments at the same condition described before (shown in Appendix A). The differential capacity curve obtained from the GCD curves (shown in Appendix A) offers a pair of peaks during the sodiation/desodiation process at 0.01 and 0.09 V, which can be ascribed to the free carbon sites. On the other hand, the peak at 0.57 V region can be assigned to the trapping of Na^+^ ions into the highly active defective graphene sites, which is irreversible in the following cycles and the formation of the SEI layer. It is thought that this peak contributes to the fluctuations in the coulombic efficiency of the SiOC electrode seen in Figure 4c. Such observations are comparable to the ones made in previous studies conducted regarding using SiOC-based materials in NIB setup [48,49].

In the case of the composite electrode fabricated with WS_2_ and SiOC material, the Na^+^ ion storage performance was also evaluated via GCD experimentation (Figure 4a). The differential capacity curve obtained (shown in Figure 4b) from the GCD plots showed a combination of peaks from the neat WS_2_ and SiOC materials. For instance, in Figure 4b, where the differential capacity curve of the WS_2_/SiOC electrode is illustrated, we observe the pair of peaks at 0.01 V and 0.08 V, which is reversible in subsequent cycles originating from the free carbon phase of the SiOC material. Three irreversible 1st cycle peaks exist at 0.3 V, 0.44 V, and 0.63 V region. While the 0.3 V and 0.63 V peaks can be assigned to the conversion reaction of the WS_2_ nanosheets and the formation of the SEI layer, the peaks seemed less intense than the neat WS_2_ material. Again, the peak residing at the 0.44 V region can be assigned to the Na^+^ ions being trapped into the defective graphene edges emerging from the SiOC material. This peak was also less intense than the neat SiOC fibermat electrodes. In addition, reversible peaks were detected at 1.36 V, 2.01 V (in sodiation steps), 2.25 V, and 1.8 V (in desodiation steps). All the peaks, as mentioned earlier, originate from the WS_2_ nanosheets within the fiber structure. The WS_2_/SiOC electrode was cycled at a gradually increasing current density of 100, 200, 400, 600, and 800 mA g^−1^ and again brought back to the initial rates in the subsequent cycles (shown in Figure 4c). The first cycle charge capacity obtained from the composite electrode was 474.88 mAh g^−1^ which is higher than the neat WS_2_ electrode (380.42 mAh g^−1^) and neat SiOC fiber electrode (141.59 mAh g^−1^). At varying current densities of 100, 200, 400, 600, and 800 mA g^−1^, the capacity of the composite electrode stood around 399.68, 313.3, 288.71, 268.1 mAh g^−1^, respectively, which is also higher than both the neat materials as well. It is observable that after harsh cycling conditions, the capacity retention at the 26th, 31st, 36th, and 41st cycles (when current densities were brought back to the original value) were 95.89%, 96.54%, 79.85%, and 68.2%, respectively which is comparable to the neat electrode. The first cycle coulombic efficiency of the WS_2_/SiOC composite electrode stood at 63.95%, similar to the neat WS_2_ powder electrode (67.21%) and higher than the neat SiOC electrode (42.33%). Interestingly, while the coulombic efficiency of the composite electrode reached nearly 100% in the 4th cycle, the coulombic efficiency of the WS_2_ powder electrode reached a similar value in the 17th cycle. For the neat SiOC electrode, it took 22 cycles.

The K^+^ ion storage mechanism of the neat WS_2_ powder electrode is evaluated in the GCD curve of Appendix A. The differential capacity curve derived from the GCD curve (shown in Appendix A) illustrates the 1st potassiation cycle peaks at 0.01, 0.35, 0.51, 0.76, and 1.25 V (vs. K/K^+^). The peak at 0.76 V (vs. K/K^+^) can be associated with the intercalation of K^+^ ions into the WS_2_ lattice to form K_x_WS_2_ [15]. This peak shifts to the 0.83 V region in the subsequent cycles. The peaks below ~0.75 V region can be ascribed to the conversion reaction from K_x_WS_2_ to K_2_S and metallic W and construction of the SEI film or other irreversible reactions, as these peaks do not appear in the subsequent cycles [10,50]. During the depotassiation process broad peak at 1.34 V (vs. K/K^+^) is observed in all the cycles signifying the deintercalation behavior of WS_2_ [51]. From the GCD curve in Appendix A, it is apparent that the 1st cycle charge capacity was 79.59 mAh g^−1^ with a very low coulombic efficiency of 24.49% at a current density value of 100 mA g^−1^. This low coulombic efficiency in the 1st cycle is thought to be happening due to the irreversible reactions taking place and also due to the ion size of the K^+^ ions, which is comparatively larger than Na^+^ ions, which should undergo much more resistance than the Na^+^ ions during the cycling process.

For this reason, the cell’s capacity is also lower than the NIB half-cell. The coulombic efficiency of the electrode reached nearly 100% after ten cycles of charge-discharge. In the rate capability test (shown in Figure 4f), the capacity retention during the 26th, 31st, 36th, and 41st cycles when the current density was brought back to the initial rate (600, 400, 200, 100 mA g^−1^, respectively) was found out to be 97.5, 92.9, 88.1 and 77.26%, respectively. The K^+^ ion storage behavior via the differential capacity curve of the SiOC material (shown in Appendix A) offered a pair of peaks at 0.01 V and 0.37 V during the potassiation/depotassiation stages. These peaks are reversible and can be attributed to the reversible K^+^ ion insertion and extraction in the free carbon domains within. A few peaks around the 0.14 V and 0.28 V region indicate the SEI layer formation, and this reaction was invisible in the following cycles [5,52].

The WS_2_/SiOC composite electrode was tested in the KIB setup (shown in Figure 4d) in the same condition as the NIB setup. The differential capacity curve obtained from the GCD experiment of the composite electrode is shown in Figure 4e. which disclosed a combination of the phenomenon of the WS_2_ neat electrode and SiOC electrode shown in Appendix A. From the differential capacity curve of the composite electrode (shown in Figure 4e), a pair of reversible peaks at 0.01 V and 0.26 V was observed, comparable to the peaks originating from the neat SiOC material during the potassiation/depotassiation stages. These peaks result from the insertion/extraction of K^+^ ions in the free carbon domain of the SiOC material validated by the spectroscopic experiments beforehand. Notably, the SEI formation due to the WS_2_ nanosheets contained within the fiber geometry can be identified by the peak at 0.51 V, which was also visible for the neat WS_2_ powder electrode that went through a slight decrease in intensity. The peak at 0.81 V, also observed in the WS_2_ neat electrode, is the result of the K^+^ ions intercalating into the layers of WS_2_ nanosheets. This peak similarly shifted in the region of 0.87 V in subsequent potassiation cycles and is indeed reversible. These peaks are observed in 1.31 V and 0.7 V regions during the depotassitation cycle, signifying the deintercalation of the K^+^ ions and is also reversible. The first cycle charge capacity obtained from the composite electrode was 218.91 mAh g^−1^ which is higher than the neat WS_2_ electrode (79.59 mAh g^−1^) and neat SiOC fiber electrode (169.66 mAh g^−1^). At varying current densities of 200, 400, 600, and 800 mA g^−1^, the capacity of the composite electrode was 158.16, 125.92, 108.41, and 96.45 mAh g^−1,^ respectively, which is also higher than both the neat materials. It is observable that after harsh cycling conditions, the capacity retention at the 26th, 31st, 36th, and 41st cycles (when current densities were brought back to the original value) were 99.6%, 98.01%, 95.39%, and 89.23%, respectively which is comparable to the neat electrode. The 1st cycle coulombic efficiency of the WS_2_/SiOC composite electrode for the KIB cell stood at 32.31%, which is higher than the neat WS_2_ powder electrode (24.49%) and the neat SiOC electrode (31.49%). This low coulombic efficiency for all the electrodes can be ascribed to the SEI layer formation of the electrode in the initial cycles. Interestingly, while the coulombic efficiency of the composite electrode was higher than the neat materials from the 2nd cycle onwards and reached near 100% at the 7th cycle, the coulombic efficiency of the WS_2_ powder electrode reached a similar value at the 12th cycle. For the neat SiOC electrode, it took 22 cycles. 

After 50 electrochemical cycles, the NIB and KIB cells containing the WS_2_/SiOC electrodes were disassembled. The spent and desodiated/depotassiated electrodes were recovered for further structural and morphological characterization. Figure 5a,b denote the SEM micrographs of the desodiated WS_2_/SiOC electrodes. Signs of microcracks in the electrodes are visible and marked with yellow dashed lined. Such microcracks are more visible at a magnified scale when presented in Figure 5b, and such phenomena can be correlated with the volume expansion of the WS_2_ nanosheets and conversion reaction. However, other significant signs of volume expansion, such as coating delamination and discoloration, were not visible in the electrodes, which signifies the structural integrity of the electrodes. On the other hand, the depotassiated WS_2_/SiOC electrodes illustrated by Figure 5c,d show very few microcracks, even on a magnified scale. Although there were no visible signs of delamination of particles and discoloration, cubicle-shaped particles were apparent from the magnified SEM micrographs (Figure 5d). These particles are considered WS_2_ nanosheets, undergoing repeated cycling processes, thus expanding their volume and forming such shapes. One can hypothesize that such cubicle shape particles do not appear in the sodiated electrodes because the radius of potassium ions is much larger than that of the sodium ions.

To summarize the overall performance, the composite electrode outperformed the neat WS_2_ nanosheet and SiOC electrodes in terms of capacity during harsh cycling conditions, capacity retention, and coulombic efficiency when individually used as Na^+^ ion and K^+^ ion half-cells. The composite electrode’s synergistic effect originates from the amorphous characteristics of the SiOC matrix not being chemically bonded to the WS_2_ nanosheets (evidenced by the spectroscopic investigations of both Raman and XPS). The SiOC material also delivers an addition of graphene-like sp^2^-bonded carbon atoms situated in the boundary of the silica tetrahedral nanodomains. This carbon also helps improve the composite electrode’s conductivity as a whole. Furthermore, the fiber geometry of this SiOC matrix adds the feature of shortening of diffusion length of ions. Additionally, the porous nature of the nanofiber configuration also provides a matrix for the WS_2_ nanosheets, suppressing the volume expansion issues experienced by the general TMD materials. This phenomenon can be coordinated with the post-cycling SEM characterization, where only a few microcracks in the cycled electrodes were visible, and no delamination and discoloration of the electrode were visible.

Furthermore, the topological defects of the free carbon phase contained within the SiOC materials and the nanovoids presented by the SiOC material provide additional sites for the alkali metal ion storage [53,54]. The lower capacity obtained from the K^+^ ion half-cell compared to the Na^+^ ion half-cell is thought to stem from the larger size of the K^+^ ions, which, when inserted in the SiOC matrix, causes repulsive interaction between the nearby K^+^ ions. Thus, in the case of the SiOC material, a much lower theoretical capacity is estimated by the previous studies (186 mAh g^−1^) [55]. However, a combination of 2D WS_2_ nanosheets helps to overcome this hurdle and raises the initial capacity to 218.91 mAh g^−1^ experimentally. Therefore, the successful combination of the two materials discussed above provided a harmonious effect that enabled the composite electrode to dominate over the neat materials when used in advanced next-generation alkali-metal ion half-cells.

## 4. Conclusions

Fabrication of SiOC/WS_2_ nanosheet composite fibermat is demonstrated via electrospinning of a preceramic polymer solution dispersed with exfoliated nanosheets followed by high-temperature pyrolysis. The composite was then investigated as a potential electrode for beyond lithium-ion electrochemical energy storage devices, i.e., Na^+^ and K^+^ batteries. Electron microscopy confirmed the dispersion of the WS_2_ nanosheets in the SiOC fiber matrix and the retention of the crystalline structure of the nanosheets even after pyrolysis at high temperatures. Spectroscopic studies further confirmed that the WS_2_ nanosheets were well protected during the fibermat pyrolysis stages. Finally, chemical functionalization of WS_2_ nanosheets in PDC fibermats in adequate amounts ensured better performance of the composite electrode when used in alkali metal ion half-cell at gradually increasing currents or harsher cycling conditions. The SiOC material provided additional sites for alkali metal ion storage and acted as a container space for volume changes induced by WS_2_. SiOC matrix may have reduced the polysulfide dissolution phenomenon in WS_2_ as the composite electrode capacity did not abruptly drop or collapse at higher cycling current densities and showed improved recovery at lower cycling currents. The capacity of the composite electrode in the KIB half-cell is lower than the NIB half-cell. The is attributed to the larger size of the K+ ion compared to the Na+ ion. Furthermore, the composite electrode in the NIB and KIB half-cells proved better capacity retention than those with neat WS_2_ and neat SiOC materials. Future investigations should be concerned with TMD materials with large interlayer spacing to accommodate K+ ions to improve cyclability.

## Figures and Tables

**Figure 1 nanomaterials-12-04185-f001:**
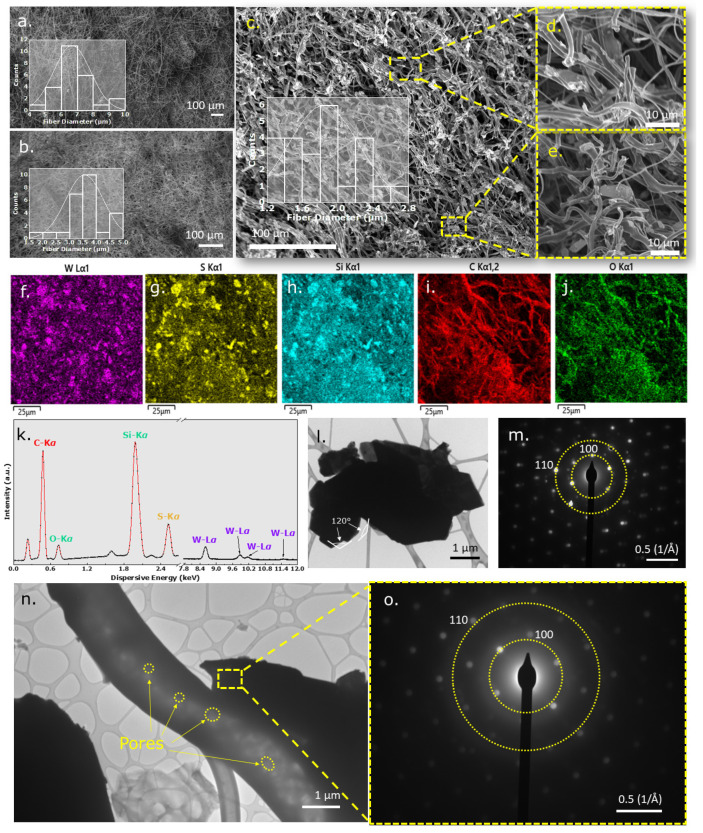
SEM micrograph of (**a**) as-spun WS_2_/SiOC fibermat confirming the fibrous structure of the fibermat; the inset showing the distribution curve of the average fiber diameter; (**b**) cross-linked WS_2_/SiOC fibermat confirming the fibrous structure of the fibermat; the inset showing the distribution curve of the average fiber diameter; (**c**) pyrolyzed WS_2_/SiOC fibermat confirming the fibrous structure of the fibermat; the inset showing the distribution curve of the fiber diameter; (**d**) magnified SEM micrograph of the pyrolyzed fibermat elucidating the individual fibers; (**e**) magnified SEM micrograph of the pyrolyzed fibermat showing WS_2_ nanosheet dispersed in the SiOC matrix; the dotted lines correspond to the parts of the fibermat from where the magnified images were taken from; (**f**) W-L*α*-1 EDS color map of the pyrolyzed fibermat confirming the presence of tungsten from the WS_2_ nanosheets on the fibermat; (**g**) S-K*α*-1 EDS color map of the pyrolyzed fibermat confirming the dispersion of sulfur in sheet like formation confirming the presence of the material from WS_2_ nanosheets; (**h**) Si-K*α*-1 EDS color map of the pyrolyzed fibermat confirming the dispersion of silicon throughout the fiber structure; (**i**) C-K*α*-1,2 EDS color map of the pyrolyzed fibermat confirming the presence of carbon on the fibermat; (**j**) O-K*α*-1 EDS color map of the pyrolyzed fibermat confirming the dispersion of oxygen throughout the fiber structure; (**k**) EDS Spectra of the WS_2_/SiOC fibermat confirming the presence of the materials within; (**l**) TEM image of the bulk WS_2_ nanosheet; (**m**) SAED pattern obtained from a part of the WS_2_ nanosheet of Figure l; (**n**) TEM image of the WS_2_/SiOC fibermat elucidating the fiber features in magnified view and confirming the dispersion of WS_2_ nanosheets within the fiber; (**o**) SAED pattern of the crystalline WS_2_ nanosheet obtained from a specific portion of the WS_2_/SiOC fibermat.

**Figure 2 nanomaterials-12-04185-f002:**
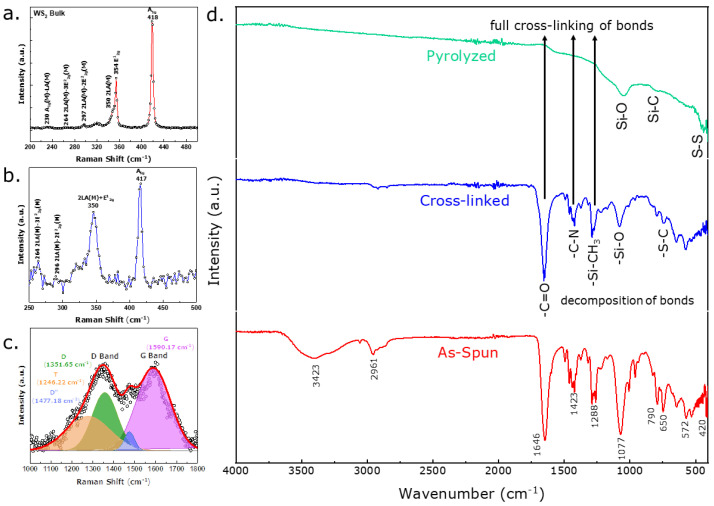
Raman Spectra of (**a**) Bulk WS_2_ powder showing the typical position of the peaks obtained from the neat material; (**b**) WS_2_/SiOC fibermat in the lower wavenumber region confirming the presence of the WS_2_ nanosheet material within the fibermat showing the same peaks as the bulk material with subtle differences; (**c**) WS_2_/SiOC fibermat in the higher wavenumber region confirming the free carbon species (integrated spectra using a gaussian function); (**d**) FTIR spectra of As-spun, cross-linked, and pyrolyzed fibermats. The spectra from top to bottom illustrate data obtained from individual pyrolyzed, cross-linked and as-spun fibermats.

**Figure 3 nanomaterials-12-04185-f003:**
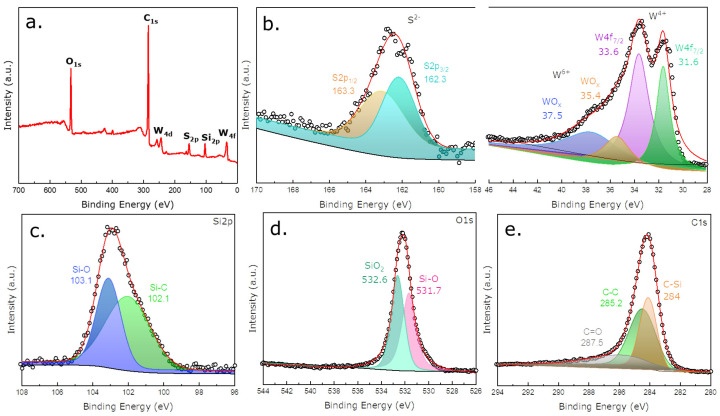
(**a**) XPS survey scan of WS_2_ containing SiOC fibermats representing the presence of individual elements after pyrolysis of the fibermat at high temperature; (**b**) High-resolution S2p (L) and W4f (R) XPS spectra of the WS_2_/SiOC fibermat recording the bonding states due to s.o.s. of W and S; High-resolution: (**c**) Si2p XPS spectra; (**d**) O1s XPS spectra; (**e**) C1s XPS spectra obtained from the WS_2_/SiOC fibermat confirming the bonding states of the Si, O and, C, respectively in SiOC matrix.

**Figure 4 nanomaterials-12-04185-f004:**
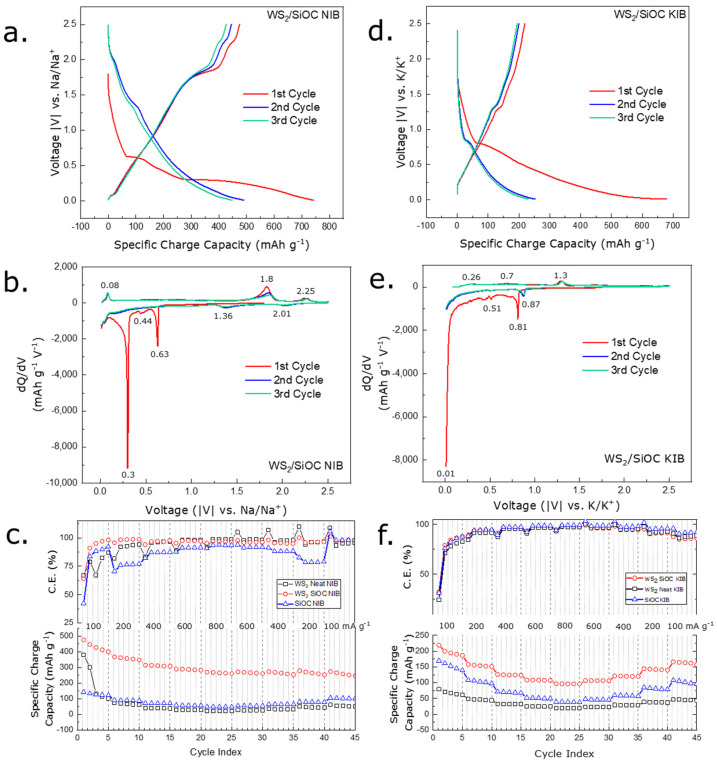
(**a**) GCD profile of the WS_2_/SiOC composite electrode when tested in a Na-ion half-cell setup; (**b**) differential capacity curve of the WS_2_/SiOC composite electrode in Na-ion half-cell setup derived from the GCD profile; (**c**) Rate capability test performed for the WS_2_/SiOC electrode when tested in a Na-ion half-cell setup at increasing current densities and bringing back to initial conditions after five cycles after each step; (**d**) GCD profile of the WS_2_/SiOC composite electrode when tested in a K-ion half-cell setup; (**e**) differential capacity curve of the WS_2_/SiOC composite electrode in K-ion half-cell setup derived from the GCD profile; (**f**) Rate capability test performed for the WS_2_/SiOC electrode when tested in a K^+^ ion half-cell setup at increasing current densities and bringing back to initial conditions after five cycles after each step.

**Figure 5 nanomaterials-12-04185-f005:**
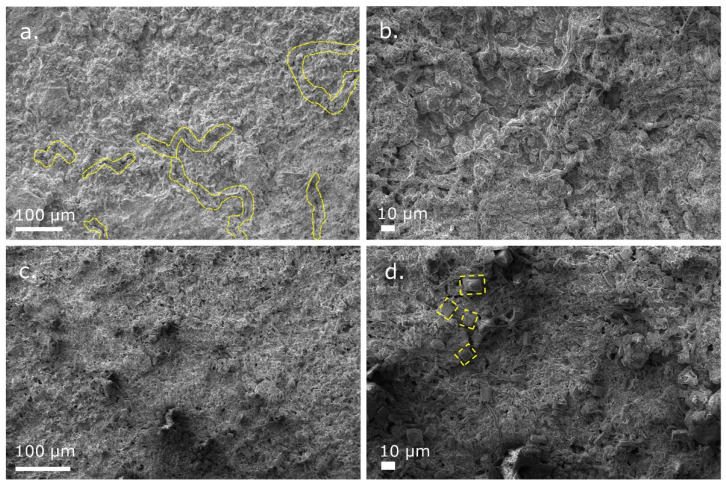
SEM micrographs of disassembled WS_2_/SiOC electrodes from (**a**) NIB half-cell; (**b**) NIB half-cell with a magnified view; (**c**) KIB half-cell; (**d**) KIB half-cell with a magnified view after 50 electrochemical cycles.

## Data Availability

The data presented in this study are available at “https://ksuemailprod-my.sharepoint.com/:f:/r/personal/sonjoy_ksu_edu/Documents/Manuscript/2022-1%20WS2%20SiOC/Data%20Availability%20WS2%20SiOC?csf=1&web=1&e=NIMqIk (accessed on 24 November 2022).”

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
