# Peer review of "WS2 Nanosheet Loaded Silicon-Oxycarbide Electrode for Sodium and Potassium Batteries"

_nanomaterials, 2022, doi:10.3390/nano12234185_

Round 1
Reviewer 1 Report
This research project successfully factionalized SiOC material with WS2 nanosheets by using electrospinning into fiber shape and annealing. The achieved structure and properties were studied and illustrated by multiple characterizations. The results were clearly presented and the conclusions were well-supported. I would like to recommend it for publication.
Author Response
Response to Reviewer 1 Comments
Comment #1: This research project successfully factionalized SiOC material with WS2 nanosheets by using electrospinning into fiber shape and annealing. The achieved structure and properties were studied and illustrated by multiple characterizations. The results were clearly presented and the conclusions were well-supported. I would like to recommend it for publication.
Author’s Response #1: The authors appreciate the reviewer’s precious time reviewing the manuscript. Necessary grammatical errors have been corrected, and changes have been incorporated in the revised version of the manuscript.
Changes Made #1: Necessary grammatical errors have been corrected throughout the manuscript.

Reviewer 2 Report
The manuscript entitled “WS2 Nanosheet Loaded Silicon-Oxycarbide Electrode for Sodium and Potassium Batteries” use the electrospinning method containing silicon-oxycarbide (SiOC) groups to functionalize WS2 to fabricate electrode for Na+ and K+ batteries. Through the mixture of WS2 and SiOC fibers, the electrochemical performance of the assembled half-cell has been improved to some extent. However, the overall electrochemical performance of the composite materials prepared by the authors is still poor, especially for the rate performance. In addition, this work also lacks innovation and highlights in the material design and preparation. In this case, I do not recommend to publish this work. Some comments are listed following:
1. In the introduction part, please clarify the energy density of 3248 mAh cm-3 for WS2 is concerning what battery type.
2. What is the purpose for the introduction of 4TTCS?
3. What are the differences between the as-spun fibers and the cross-linked fibers?
4. Please arrange the figures in the supplementary materials according to the sequence in which they appear in the main text.
5. The caption of the figures in the main text are too long and should be simplified.
6. In Figure 1i, how does the hole structure generate? Why these holes can be observed in TEM but not in SEM images?
7. Also in Figure 1i, it seems that the fibers are mechanically mixed with WS2. How the authors ensure the stability of the composites during cycling?
8. From Figure 4c, it is noticed that the columbic efficiency exhibits violent fluctuations during the rate capacity testing process. Please explain such phenomenon.
9. The authors claimed that “the porous nature of the nanofiber configuration also provides a matrix for the WS2 nanosheets which suppresses the volume expansion issues experienced by the general TMD materials”. However, the TEM image provided by the authors exhibit a simple mechanical mixing of SiOC matrix and WS2 sheets. How does this structure suppress the volume expansion?
Reviewer 3 Report
The authors report the successful synthesis of the successful functionalization of SiOC material with WS2 nanosheets for ultrahigh capacity and rate capability Na/K ion batteries. As a result, this WS2/SiOC composite delivers high capacity, good rate capability and cycling stability for both SIBs and PIBs. The results were interesting and meaningful. Minor revision should be addressed before it can be accepted. Following comments and questions are intended for the authors to improve their work.
1. There are some grammatical or spelling errors. The language of the manuscript should be improved.
2. The mass loading of the active materials in the electrodes should be specified.
3. The characterization after cycles should be tested to prove the superior property of WS2/SiOC electrode.
4. The initial coulombic efficiency of WS2/SiOC composite for PIBs seems too low, please explain. Besides, why the CV curves are not close?
5. For presentation the advances of WS2/SiOC electrode materials, the comparison of battery properties of WS2/SiOC with previous reports of similar materials should be provided.
6. To interpret the outstanding battery properties of the WS2/SiOC materials, other kinds of metal sulfides electrode materials for Na+/K+ batteries are helpful, see: Chem. Eng. J. 451 (2023) 138508.; Coord. Chem. Rev. 464 (2022) 214544
